# Burden of scabies in a Ghanaian penitentiary

Yaw Ampem Amoako[1,2,3]*, Michael Ntiamoah Oppong[1], Dennis Odai Laryea[4], Gloria Kyem[1], Solomon Gyabaah[1], Abigail Agbanyo[1], Bernadette Agbavor[1], Nana Konama Kotey[5], Irene Dzathor[6], Felicia Owusu-Antwi[7], Kingsley Asiedu[8], Richard Odame Phillips[1,3], Ymkje Stienstra[2,9]

1 Kumasi Centre for Collaborative Research in Tropical Medicine, Kwame Nkrumah University of Science and Technology, Kumasi, Ghana, 2 Department of Internal Medicine/Infectious Diseases, University Medical Center, Groningen, University of Groningen, Groningen, Netherlands, 3 School of Medicine and Dentistry, Kwame Nkrumah University of Science and Technology, Kumasi, Ghana, 4 Disease Surveillance Department, Ghana Health Service Headquarters, Accra, Ghana, 5 Neglected Tropical Diseases Control Programme, Ghana Health Service Headquarters, Accra, Ghana, 6 FHI 360 Country Office, Accra, Ghana, 7 World Health Organization Country Office, Accra, Ghana, 8 World Health Organization, Geneva, Switzerland, 9 Department of Clinical Sciences, Liverpool School of Tropical Medicine, Liverpool, United Kingdom

* yamoako2002@yahoo.co.uk

## Abstract

### Background

There is a dearth of information concerning the epidemiology of human scabies in prisons. Therefore, we aimed to assess the burden of scabies and ascertain if prevalence was high enough to warrant mass drug administration (MDA) with ivermectin in a medium security prison in central Ghana.

### Methods

We conducted a cross-sectional study in December 2022 and recruited inmates at the Kumasi central prison in Ghana. Medical history and demographic information was collected using a REDCap questionnaire. A standardised skin examination of exposed regions of the body was performed on all participants and scabies was diagnosed based on the criteria of the International Alliance for the Control of Scabies (IACS).

### Results

Of the 559 participants, 19 (3.4%) were female. The median (IQR) age was 36 (33–40) years. There were 368 cases (65.8%) of scabies which was mostly located on the hands, fingers and finger webs. No female inmate had scabies. Scabies severity was predominantly mild (63.3%) or moderate (30.7%). Among males, the median (IQR) number of persons per cell was 115 (56–118) and 7 (7–8) for female prisoners. 79.2% of 96 individuals previously treated in the preceding two months still demonstrated skin manifestations of scabies. Impetigo was found in 12.9% of participants. Seventeen percent of participants with scabies had impetigo compared to 5.8% in individuals without scabies [RR 2.9 (95% CI 1.6–5.5)].

**Data Availability Statement:** All relevant data are within the manuscript.

**Funding:** YS received a grant from the NWA Idea Generator (grant number NWA.1228.192.144). The

funders had no role in study design, data collection and analysis, decision to publish, or preparation of the manuscript.

**Competing interests:** The authors have declared that no competing interests exist

## Conclusion

A very high proportion of inmates suffered from scabies in the prison. MDA with ivermectin and health education are needed to reduce the burden of scabies in the prison. Its implementation and effectiveness should be studied.

## Background

Inmates of prisons have significant healthcare needs. Needs range from communicable diseases to mental health and dental care [1–3]. The communicable diseases challenging the health of prisoners include pneumonia, tuberculosis, diarrhoea, hepatitis and skin conditions such as scabies [4–7].

Scabies is a highly contagious parasitic infestation of the skin caused by the mite *Sarcoptes scabiei var. hominis* and is characterized by intense pruritus, redness, and rash. Scabies is a global health problem, affecting millions of people every year, and it is particularly prevalent in resource-limited settings, including prisons [8, 9]. The overcrowded conditions within prisons can facilitate scabies transmission. Close contact between inmates during activities such as sleeping, sharing bedding or clothing, and participating in recreational activities can facilitate the spread of scabies mites through direct contact. Additionally, factors such as inadequate access to clean water and soap and limited laundry facilities can further exacerbate the spread of scabies in prisons. The capacity of health services in low middle income countries to detect and treat cases of scabies is also limited largely as a result of low awareness and under-diagnosis [8, 10]. Even when cases are identified, there are delays in treatment due in part to non-availability of drugs or high cost of treatment especially for persons without health insurance [11]. In the prison setting, a failure to adequately screen and manage scabies in incoming prisoners, as well as inadequate training of staff on scabies prevention and control practices and a lack of effective scabies control programme, can exacerbate the problem. Studies have shown that the prevalence of scabies vary widely, ranging from 2% to 70%, depending on the location/ region, living conditions, and characteristics of the population studied [9, 12–14]. For example, a study conducted in a prison in Poland found a scabies prevalence rate of 2.3% among inmates [12]. In Tanzania, a much higher prevalence of 69.5% has been reported [13].

The risk of scabies outbreaks is increased in prisoners due to various factors, such as limited access to healthcare, inadequate treatment of cases, lack of awareness among prisoners and staff. Furthermore, delayed diagnosis and treatment of scabies, promotes the spread of the infection among inmates and staff resulting in increased morbidity (especially in cases with secondary infections), mental health distress and increased healthcare costs [15, 16].

Scabies infestation can also cause secondary skin infections, such as impetigo and cellulitis due to scratching and skin breakdown, which can be exacerbated in a prison setting where hygiene practices may be compromised.

Following the categorisation of scabies as a Neglected Tropical Disease (NTD), measures to effectively control the scourge of the disease have been proposed. Mass Drug Administration (MDA) with ivermectin has emerged as one such promising tool for scabies control in endemic settings [15, 16]. MDA with ivermectin leads to a substantial decrease in prevalence of scabies with an additional benefit of about 90% relative reduction in the prevalence of impetigo [15].

Several studies have reported on scabies in community and school [17, 18] and hospital [19, 20] settings in Ghana. However, there is a dearth of information concerning the epidemiology

of human scabies in Ghanaian prisons. Therefore, we aimed to assess the burden of scabies and ascertain if its prevalence was alarmingly high enough to warrant MDA with ivermectin in the prison.

## Methods

### Ethics statement

Written informed consent was obtained from all participants. Written permission was also obtained from the authorities of the Kumasi Central prisons. Ethical approval for the study was granted by the Committee on Human Research, Publications and Ethics (CHRPE) of the School of Medicine and Dentistry of the Kwame Nkrumah University of Science and Technology in Ghana (approval number: CHRPE/AP/037/21, CHRPE/AP/829/22 and CHRPE/AP/ 1060/23). Participant autonomy, confidentiality, and welfare were always maintained and given the highest priority. At any time, study participants had the option of requesting withdrawal and removal of their coded data from the study without it affecting their treatment. Both inmates participating in the study diagnosed with scabies and inmates not participating in study reporting to study team with scabies symptoms were treated. All study procedures conformed with the principles guiding research in human subjects as set out in the Declaration of Helsinki [21].

### Study setting

This cross-sectional study was conducted among inmates of the Kumasi Central Prison, a medium security prison located in Kumasi, the capital of the Ashanti Region and the second largest city in Ghana. The total number of inmates as of 16th December 2022 was 1,906 made up of 1,886 males and 20 females. Males and females are kept in separate areas of the prisons complex.

The prison has a 23- bed infirmary. The infirmary is manned by a trained Physician assistant with support provided by other staff of the prison service. Prisoners are placed in cells within blocks A, B and C. The A block has 8 cells (A1-A8) each of which houses an average of 110 inmates. The remaining male prisoners are housed in dormitories in blocks B and C. The females are housed in a single dormitory where each of them has a separate bed. While some inmates have access to bunk beds, more than half of all inmates sleep on the floor. In the prison, inmates are allowed some free time outside their cells to interact and engage in play, walks or have meals within a common zone/ space for limited periods during the day. Inmates also interact with staff and medical personnel during the period.

### Participant recruitment and study procedures

From 19th to 23rd December 2022, inmates were contacted within their cells by members of the study team and informed about the assessment and consent sought. A convenience sampling technique was employed to identify and select participants for the study. We aimed to include one third of all inmates in the study. To achieve this, inmates from each cell or dormitory within the prison were made to stand in line and every third prisoner who provided informed consent was enrolled. Individuals who did not provide informed consent were excluded from the study. All examinations were performed within the consulting area of the prison's infirmary. The consulting area of the infirmary consisted of a space which was divided into 3 cubicles with screens.

The skin examinations were conducted by medical doctors with clinical experience diagnosing scabies based on earlier activities in infectious diseases and/or public health. The

research team received supplemental training on the diagnosis of scabies, impetigo and other locally common skin conditions as well as the use of the International Alliance for Control of Scabies (IACS) criteria for scabies diagnosis; this was provided to team members by the lead researcher (YAA).

The demographic details were recorded with the aid of a REDCap based questionnaire. With the individual participant in short sleeved shirts and shorts, clinical examination of the exposed skin areas was performed using a standardized approach as previously reported [17, 22]. Shoes were removed prior to examination. The breasts and genitals were not examined, except when requested by participants and then only in a separate, private examination area. To ensure confidentiality, history taking and clinical examinations were conducted in a private space within the clinic.

A focused history of standardized questions was taken of all participants comprising of information required for the IACS categorisation. Questions included whether participants experienced itch. Contact history was assessed by asking if participants had a cell mate with itch or others with itch before being imprisoned, or a cell mate or with others with scabies rash before being imprisoned. Participants were shown images of people with typical scabies rashes to assist these questions.

Questions on treatment included whether participants had received any scabies treatment in the preceding months, what treatment was received if any and a description of how the treatment was used. History was taken in the local language (with the assistance of interpreters where required) or in English.

The diagnosis of scabies was based on the B1, B3, C1 and/or C2 criteria developed by the IACS [23]. The study team also looked for crusted scabies. Impetigo was diagnosed based on the presence of papules, pustules or ulcerative lesions with associated erythema, crusting or pus. Using previously published criteria [24], the severity of scabies and impetigo were categorised based on the number of lesions present as: mild (1 to 10 lesions), moderate (11 to 49 lesions) or severe (50 or more lesions). Impetigo was classified as: very mild (1 to 5 lesions), mild (6 to 10 lesions), moderate (11 to 49 lesions) or severe (50 or more lesions).

All individuals with scabies were treated with benzyl benzoate as recommended by the Ghana National Treatment guidelines [25]. Participants with impetigo were also treated as per standard protocol in Ghana [25].

## Statistical analysis

Data in RedCap was extracted into Microsoft Excel version 2013 (Microsoft Corp., USA) and analysed using Epi Info version 7.2.2.2 and Stata statistical software version 13 (StataCorp LLC, USA). Categorical variables were expressed as frequencies and proportions; and results for continuous variables were expressed as median and interquartile range (IQR). Categorical data were analyzed using the Chi square test of association and the Fisher's exact test. Medians were compared using the Mann-Whitney test. The Relative Risk (RR) of impetigo in participants with or without scabies was calculated with 95% confidence interval (CI). All statistical tests were performed at a 95% confidence level and a p value $<0.05$ was deemed statistically significant.

## Results

Overall, 559 participants were recruited. Nineteen of the 20 female prisoners consented to participate and no male prisoners refused participation. Most participants were male (540/559 or 96.6%), and the median age was 36.0 (IQR 33–40) years.

**Table 1. Characteristics of participants with and without scabies.**

| | Participants with scabies, n = 368 | Participants without scabies, n = 191 | All, n = 559 | p value |
|---|---|---|---|---|
| Itch (%) | 368 (100.0) | 137 (71.73) | 505 (90.3) | 0.0001 |
| Rash (%) | 364 (98.9) | 97 (50.78) | 461 (82.5) | 0.0001 |
| Contact positive (%) | 368 (100.0) | 159 (83.2) | 527 (94.3%) | 0.0007 |
| Number of people per cell, median (IQR) | 115 (57–118) | 108 (17–117) | 112 (55–118) | 0.0001 |
| Time (median/IQR) spent within cell (hours/day) | 14 (14–15) | 14 (14–16) | 14 (14–15) | 0.39 |
| Duration of stay (median/IQR) in the prison (months) | 6 (3–12) | 8 (4–31) | 7 (3–18) | 0.15 |

Three hundred and sixty-eight (368) of the 559 (65.8%) participants were found to have clinical scabies as assessed by the IACS criteria (Table 1). Scabies severity was mild in 63.3% and moderate in 30.7% of affected individuals. Scabies lesions were most common on the hands, wrists, fingers and finger webs. None of the affected individuals had crusted scabies. Seventy-two (72/559 or 12.9%) had impetigo which was mostly very mild (45.9%) or mild (44.4%) in severity (Table 2).

No female inmate had clinical scabies. Ninety one percent (507/559) of participants reported that scabies was a common problem in the prison. None of the female prisoners perceived scabies as a problem in their prison.

Contact history was positive in 100% of participants with scabies. Fifty seven percent (57%) had burrows and 98.9% of participants with clinical scabies had rash typical for scabies (Table 1). All participants with scabies reported itch (with a median (IQR) duration of 30 (14–60) days. Only 6% of participants reported no scabies contact in the past weeks. Further, (490/540) 90.7% of male participants reported sharing a bed or bed linen with other cell mates; no female participant shared a bed or linen.

**Table 2. Scabies and impetigo severity in participants with or without clinical scabies.**

| | | Clinical scabies, not treated | Clinical scabies, previously treated | No clinical scabies, not treated | No clinical scabies, previously treated |
|---|---|---|---|---|---|
| Total (n = 559) | | 292 | 76 | 171 | 20 |
| IACS category | B1 (%) | 207 (70.9) | 2 (2.6) | NA | NA |
| | B3 (%) | 67 (22.9) | 50 (65.8) | | |
| | C1 (%) | 9 (3.1) | 13 (17.1) | | |
| | C2 (%) | 9 (3.1) | 11 (14.4) | | |
| Positive contact history (%) | | 292 (100.0) | 76 (100.0) | 140 (81.9) | 19 (95.0) |
| Scabies severity | | | | | |
| Mild (%) | | 192 (65.8) | 41 (54.0) | NA | NA |
| Moderate (%) | | 87 (29.8) | 26 (34.2) | | |
| Severe (%) | | 5 (1.7) | 5 (6.6) | | |
| Missing information (%) | | 8 (2.7) | 4 (5.3) | | |
| Impetigo severity (n = 72) | | | | | |
| Very mild (%) | | 21 (45.7) | 7 (46.7) | 3 (33.3) | 2 (100.0) |
| Mild (%) | | 20 (43.4) | 7 (46.7) | 5 (55.6) | 0 (0) |
| Moderate (%) | | 5 (10.9) | 1 (6.6) | 1 (11.1) | 0 (0) |
| Severe (%) | | 0 (0) | 0 (0) | 0 (0) | 0 (0) |

*IACS = International Alliance for Control of Scabies

The median (IQR) number of people per cell in the prison was 112 (55–118) as reported by the participants. Among males the median was 115 (56–118) and 7 (7–8) for female prisoners. The median duration of stay in the prison was 7 (3–18) months. Prisoners tended to spend a median of 14 (14–15) hours in their cell.

At the time of assessment, 96 (20.7%) of the participants with scabies had recently received treatment in the past two months because of their skin problems. Ten of these reported receiving topical Benzyl benzoate while 48 received oral ivermectin, 1 received topical permethrin and 37 were treated with other medications such as sulphur and Whitfield's ointment. Seventy-six (79.2%) of the 96 individuals previously treated still demonstrated skin manifestations which led to an IACS scabies diagnosis. Twenty of the inmates previously treated were free of scabies at the time of skin assessment. In participants without scabies, itch (71.7%) and rash (50.8% were common but these were less than in those with scabies (100% and 98.9% respectively) as shown in Table 1. Seventeen percent of participants with scabies had impetigo compared to 5.8% in individuals without scabies [RR 2.9 (95% CI 1.6–5.5)].

## Other skin conditions found

Some inmates were found to have tinea corporis (39), tinea capitis (7), pityriasis versicolor (12), eczema (16), skin ulcer (7), abscess (7), lichen planus (2), and Kaposi sarcoma (1). Inmates with these conditions were treated on spot or referred for further care.

## Discussion

Scabies is an infectious parasitic disease that has the potential to spread rapidly in confinement conditions as exists in prisons. In this study in a Ghanaian prison, 65.8% of inmates who were examined had clinical scabies which was mainly mild to moderate in severity. All affected individuals were male with median age of 36 years. Further, impetigo was common among individuals with scabies (17%) than in those without scabies (5.8%). We also identified fungal dermatosis, ulcers, abscess and Kaposi sarcoma in the prison population.

The higher prevalence of scabies in this confinement institution may be as a result of the overcrowding of the prison leading to a large number of prisoners per cell. Furthermore, close contact between prisoners during sleep times may promote spread of scabies.

In Cameroon, a scabies prevalence of 32% has been reported among prisoners and overcrowding, male gender, sharing of clothing and low educational status, were independent drivers of scabies occurrence in the three prisons studied [9]. Leppard and colleagues [13] found a prevalence of 69.5% in a Tanzanian prison. In another study, a scabies prevalence of 2.3% was found among Polish prisoners [12]. In some countries [12], initial medical examinations are carried out by a general practitioner within a few days after imprisonment, while diagnostic examinations are performed at intervals; such scheduled assessments can lead to early detection and treatment of scabies cases and reduce the potential for outbreaks. The overcrowding and lack of systematic efforts to detect and control scabies in the Ghanaian prison may account for the very high scabies prevalence observed in the current study. It is recommended that the prison institutes a scabies control programme that incorporates health education and scheduled or periodic examination of inmates to promote early detection and appropriate treatment of scabies and other common skin conditions to facilitate control. Additionally, inmates should be encouraged to report their own disease to the prison service, as this can substantially accelerate the implementation of therapy for scabies. Efforts should be made to decongest the prison and improve the general access to healthcare in the prison. Health education for inmates and staff can also help reduce stigmatization of affected individuals in such settings.

Mental disorders like depression and anxiety are prevalent among Ghanian prisoners [26]. Intense pruritus from scabies can further threaten mental health and may lead to increased tension between inmates.

In our study, 17% of participants had received treatment for their skin problems in the t 2 months preceding the assessment. However, 79.2% of the individuals who were previously treated still exhibited skin manifestations of scabies; these may include individuals with post scabies itch, reinfections/ recurrences or non-response to treatment/ treatment failure. There had been no treatment of the contacts of previously identified scabies cases. The lack of a systematic approach to the treatment of previously identified scabies cases and their contacts likely led to reinfection of individuals and ongoing spread within the prison. Additionally, the overcrowded living conditions, limited access to healthcare and inadequate Water, Sanitation and Hygiene (WASH) facilities may have facilitated scabies spread within the prison. There is the need to explore strategies for more effective control of scabies within this prison setting. A high percentage of individuals without scabies had an itch or rash but did not meet the IACS criteria for clinical scabies. It is possible that the IACS criteria does not work as well in such high-risk settings. These may also include persons with alternative diagnosis such as fungal dermatosis, eczema, pediculosis or other infestations.

There are different strategies for scabies control. A control strategy of treating clinically affected persons and their contacts may provide temporary relief for the individuals but has limited success in reducing population prevalence in the longer term [27].

Mass drug administration (MDA) using topical permethrin or oral ivermectin offers an alternative approach for population control to substantially reduce the burden of scabies.

Mass treatment of highly endemic communities with topical 5% permethrin has been reported to substantially reduce scabies prevalence [28–31]. In another study with permethrin, although the scabies prevalence remained unchanged, the prevalence of secondary infected scabies decreased from 3.7% to 1.5% indicating a relative reduction of 59% [32].

Topical permethrin is expensive and requires application to the whole body. Other disadvantages of topical agents include inadequate application to local lesions, a lack of privacy to undertake whole-body application, insufficient facilities to rinse off the cream, poor compliance among contacts of scabies cases and discomfort due to topical treatment [33]. These factors constitute major barriers for effective use of topical therapy for population control of scabies, especially in the prison setting.

Ivermectin, as an oral therapy can offset some of the challenges associated with topical treatment and can potentially increase compliance resulting in better control of scabies in affected populations [8, 34]. Studies on MDA with ivermectin performed in island settings [15, 16, 35] reported a reduced prevalence of scabies. Compared to topical permethrin, oral ivermectin resulted in larger reductions in scabies prevalence at 12 and 24 months [15]. Recently, mass screening and community treatment (MSAT) of cases and their contacts with oral ivermectin delivered by community health workers has been shown to be superior to mass screening followed by usual care involving referral to clinic for topical treatment in controlling scabies in India [34]. While MSAT with ivermectin showed promising results and has relatively lower costs, the approach might miss persons with subclinical scabies making it less effective than MDA [34]. Furthermore, MSAT does not target people who work in the prison and may be contacts of scabies cases.

We found a scabies prevalence in the prison which is much higher than the 10% threshold set for MDA [36]. In the current study, individuals with scabies were treated with topical benzyl benzoate as recommended in Ghana. However, the high prevalence and large number of contacts (entire prison population including staff and health workers) make MDA with ivermectin imperative. An urgent collaborative effort involving multiple stakeholders such as the

Ghana Prisons Service, the Ghana Health Service, civil society and non-governmental organisations and donors is needed to mobilise adequate doses of ivermectin to undertake an MDA in this institutionalized setting.

Ivermectin is contraindicated in children < 5 years old, persons with severe concurrent illness, individuals with hypersensitivity to components of the drug, and pregnancy. Inadequate community engagement, lack of political support and high costs are potential barriers that may pose challenges for MDA implementation [37, 38]. Furthermore, due to its lack of ovicidal action, a second dose of ivermectin is usually recommended after 7–14 days for individual treatment, to kill newly hatched mites. Administration of a second dose that is separated by several days as MDA is logistically challenging and is associated with higher costs. An oral agent with a longer duration of activity in the skin, that could persist for sufficient time to kill newly hatched mites, would make the requirement for a second dose unnecessary. Moxidectin with a half-life of up to 43 days, and prolonged activity in the skin offers promise in this regard [39]. An initial preclinical study reported that oral single dose moxidectin was more effective than two consecutive doses of ivermectin for treating scabies [40]. In future moxidectin may be even more helpful as its longer half-life may render it more efficient in the prison setting where interactions between people are not restricted by geographic boundaries as exists in island populations.

## Study strengths and limitations

Our study included a relatively large number of individuals and is the first systematic assessment of the burden of scabies among prisoners in Ghana. The study was conducted in the second largest prison in the country and similar outbreaks may be ongoing in other prisons in Ghana. We used the IACS criteria for standardisation of scabies diagnosis and this was easily and effectively applied in this resource limited setting in Ghana. However, there are also certain limitations. The diagnosis of scabies burrows was not confirmed with dermoscopy. However, a systematic review of diagnostic methods of scabies has shown low accuracy of dermoscopy in diagnosing scabies and clinical diagnosis remains an accepted practice in field studies [41]. In addition, dermoscopy is not readily available in most resource limited settings. Furthermore, the IACS criteria have not been validated in the prison setting where there may be a different clinical background of skin problems than the settings the IACS criteria was validated in. We did not perform molecular based tests like PCR nor use tools like cultures for laboratory confirmation of fungal or bacterial infections. No skin biopsies were taken for histological examination. These tests are expensive and not routinely available to the population in which the current study was conducted.

## Conclusion

There is a very high prevalence of scabies and impetigo in this medium security Ghanaian prison. Monitoring the health of prisoners can facilitate early response to emerging cases of scabies and help limit the potential for outbreaks in prisons. To minimise the problem, urgent measures including health education and MDA with ivermectin are needed to address this major public health issue in the prison.

## Acknowledgments

We would like to thank the administration and staff of the Kumasi Central prisons. We appreciate all the inmates of the prison for their practical involvement in this study. A special thanks goes to Mr Collins Asare and staff of the skin NTDs Research group at the Kumasi Centre for Collaborative Research for providing administrative support.

## Author Contributions

**Conceptualization:** Yaw Ampem Amoako, Richard Odame Phillips, Ymkje Stienstra.

**Data curation:** Yaw Ampem Amoako, Michael Ntiamoah Oppong.

**Formal analysis:** Yaw Ampem Amoako, Dennis Odai Laryea.

**Investigation:** Yaw Ampem Amoako, Michael Ntiamoah Oppong, Gloria Kyem, Solomon Gyabaah, Abigail Agbanyo, Bernadette Agbavor.

**Methodology:** Yaw Ampem Amoako, Richard Odame Phillips, Ymkje Stienstra.

**Project administration:** Yaw Ampem Amoako, Abigail Agbanyo, Bernadette Agbavor.

**Resources:** Dennis Odai Laryea, Nana Konama Kotey, Felicia Owusu-Antwi, Kingsley Asiedu, Ymkje Stienstra.

**Supervision:** Richard Odame Phillips, Ymkje Stienstra.

**Validation:** Dennis Odai Laryea, Gloria Kyem, Solomon Gyabaah, Abigail Agbanyo, Bernadette Agbavor, Nana Konama Kotey, Irene Dzathor, Felicia Owusu-Antwi, Kingsley Asiedu, Richard Odame Phillips, Ymkje Stienstra.

**Visualization:** Michael Ntiamoah Oppong, Dennis Odai Laryea, Gloria Kyem, Solomon Gyabaah, Abigail Agbanyo, Bernadette Agbavor, Nana Konama Kotey, Irene Dzathor, Felicia Owusu-Antwi, Kingsley Asiedu, Richard Odame Phillips.

**Writing – original draft:** Yaw Ampem Amoako, Ymkje Stienstra.

**Writing – review & editing:** Yaw Ampem Amoako, Michael Ntiamoah Oppong, Dennis Odai Laryea, Gloria Kyem, Solomon Gyabaah, Abigail Agbanyo, Bernadette Agbavor, Nana Konama Kotey, Irene Dzathor, Felicia Owusu-Antwi, Kingsley Asiedu, Richard Odame Phillips, Ymkje Stienstra.

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
