## [Decision Letter · Decision Letter 0]

24 May 2024

PONE-D-24-01158Burden of scabies in a Ghanaian penitentiaryPLOS ONE

Dear Dr. Amoako,

Thank you for submitting your manuscript to PLOS ONE. After careful consideration, we feel that it has merit but does not fully meet PLOS ONE’s publication criteria as it currently stands. Therefore, we invite you to submit a revised version of the manuscript that addresses the points raised during the review process. Kindly address the revisions requested by the reviewers.

We look forward to receiving your revised manuscript.

Kind regards,

Henok Dagne

Academic Editor

PLOS ONE

Journal Requirements:

2. Please provide additional information regarding the considerations  made for the prisoners included in this study. For instance, please discuss whether participants were able to opt out of the study and whether individuals who did not participate receive the same treatment offered to participants.

3. We note that your Data Availability Statement is currently as follows: [All relevant data are within the manuscript and its Supporting files]

4. We note that Figure 1 in your submission contain copyrighted images. All PLOS content is published under the Creative Commons Attribution License (CC BY 4.0), which means that the manuscript, images, and Supporting Information files will be freely available online, and any third party is permitted to access, download, copy, distribute, and use these materials in any way, even commercially, with proper attribution. For more information, see our copyright guidelines: http://journals.plos.org/plosone/s/licenses-and-copyright.

Reviewers' comments:

Reviewer's Responses to Questions

**Comments to the Author**

1. Is the manuscript technically sound, and do the data support the conclusions?

Reviewer #1: Partly

Reviewer #2: Partly

2. Has the statistical analysis been performed appropriately and rigorously? 

Reviewer #1: No

Reviewer #2: Yes

3. Have the authors made all data underlying the findings in their manuscript fully available?

Reviewer #1: No

Reviewer #2: Yes

4. Is the manuscript presented in an intelligible fashion and written in standard English?

Reviewer #1: Yes

Reviewer #2: No

5. Review Comments to the Author

Reviewer #1: Congratulations to the authors for coming up with this manuscript. As yet, there seems not to be any data on this topical issue in Ghanaian prisons. The Document is well written but will require some further corrections.

Abstract:

1. For the statement “and ascertain if the prevalence was high enough to warrant mass drug administration (MDA) with ivermectin…”, Consider deleting this as it is not an objective but a derived conclusion. Note that this study does not determine if MDA should be done.

2. “79.2% of 96 individuals previously treated still demonstrated skin manifestations of scabies.”. When was this previous treatment? Was it before the study or a follow-up after the study? This must be indicated.

3. “as it is likely that the criteria to start MDA for scabies are regularly met in prisons worldwide”: What is the basis for this statement?

Background

1. “and ascertain if its prevalence was alarmingly high enough to warrant MDA with ivermectin in the prison.” : How is this to be determined?

Methods

1. “Every third prisoner within a cell who provided informed consent was enrolled”: As scabies is greatly determined by the space and environment one lives in, the sampling technique should be appropriately described to enable readers to determine if it is representative enough of the prison population.

Results

1. Table 2 should rather come first. The contents of Table 1 are not mentioned in the first two paragraphs, after which Table 2 is mentioned.

2. The entirety of the contents of Table 3 is also in Table 1, and thus just a repetition. It should be deleted.

3. The norm is that p-values should be exact if >=0.001 before cutting them short as < .001. I suggest you do just that to these p-values. A p-value of 0.04 is 10x that of 0.004!

4. In Table I for the rows: “Number of people per cell, median (IQR)” and “Time (median/IQR)”: There was no valid test to compare medians in the statistical analysis section, Is that what is being done here? If so, please put that in the statistical analysis section as well.

5. Why is the p-value for the “Duration of stay (median/IQR) in the prison (months)” blank? Should it not be filled in?

6. Abbreviations such as “IACS” in the tables should be explained in the legend.

Reviewer #2: Major Comments:

1. Authors reported the scabies prevalence of 65.8% among inmates which is significantly higher. Authors are suggested not rely only on clinical diagnosis but also prefer confirmatory tests like dermoscopy, culturing, PCR etc. especially in such a high prevalence rate.

2. Authors reported that a significant percentage (79.2%) of previously treated individuals still showed skin manifestations of scabies. This raises questions about treatment efficacy and possibility of reinfection and its potential factors. Authors are suggested to discuss the potential reasons behind this observation/treatment failure. It would be appropriate to include the exploring strategies for more effective treatment.

3. Authors stated the (MDA with ivermectin) as a potential solution, authors are suggested to include the potential drawbacks of this approach. Authors may add alternative combination approaches for scabies control.

4. There are already similar studies have been published. Therefore, authors should justify the objectives and novelty of the current study.

5. What were the inclusion and exclusion criteria? It would be more appropriate to add sample size calculation and its justification? As sample size is vital to represent the study population.

6. Explain if the questionnaire was pilot tested. Have the participants completed an informed consent? It remains to include the study population, the sample and the sampling.

7. Authors should describe limitations of their study. Furthermore, these limitations should be discussed.

Additional Comments:

1. Proof read the article to improve sentence structure.

2. Read similar articles to polish your write-up. e.g. https://www.mdpi.com/1648-9144/59/11/1905. This article may help to improve your understandings and quality of article.

6. PLOS authors have the option to publish the peer review history of their article (what does this mean?). If published, this will include your full peer review and any attached files.

Reviewer #1: **Yes: **Samuel Blay Nguah

Reviewer #2: **Yes: **Mehmood Ahmad

---

## [Author Response · Author response to Decision Letter 0]

1 Jun 2024

31st May, 2024

Kumasi Centre for Collaborative Research

Kwame Nkrumah University of Science and Technology 

Kumasi, Ghana

The Editor

PLOS One Journal

Dear Sir,

Re: ‘Burden of scabies in a Ghanaian penitentiary’ [PONE-D-24-01158] - [EMID:8ffb31d56adc6509]

My co-authors and I have taken note of the review comments and revised the manuscript as suggested. 

We wish to submit the revised manuscript for publication in your esteemed journal. In the attached reply, the reviewer questions are in blue font and our responses are in red font. 

We look forward to the next steps towards the publication of our manuscript.

Yours Sincerely,

Dr Yaw Ampem Amoako

Corresponding author

Point-by-point response to review comments

Journal Requirements:

Response: Done. The manuscript has been formatted according to the journal’s style.

2. Please provide additional information regarding the considerations made for the prisoners included in this study. For instance, please discuss whether participants were able to opt out of the study and whether individuals who did not participate receive the same treatment offered to participants.

Response: We have included the following sentences in the ethics section to address the editor’s query: ‘Participant autonomy, confidentiality, and welfare were always maintained and given the highest priority. At any time, study participants had the option of requesting withdrawal and removal of their coded data from the study without it affecting their treatment. Both inmates participating in the study diagnosed with scabies and inmates not participating in study reporting to study team with scabies symptoms were treated.’ (line 123-127). 

3. We note that your Data Availability Statement is currently as follows: [All relevant data are within the manuscript and its Supporting files]

Response: All relevant data are within the manuscript.

4. We note that Figure 1 in your submission contain copyrighted images. All PLOS content is published under the Creative Commons Attribution License (CC BY 4.0), which means that the manuscript, images, and Supporting Information files will be freely available online, and any third party is permitted to access, download, copy, distribute, and use these materials in any way, even commercially, with proper attribution. For more information, see our copyright guidelines: http://journals.plos.org/plosone/s/licenses-and-copyright.

Response: The lead author is member of the International Alliance for Control of Scabies (IACS). The photo in figure 1 might have been part of those submitted to IACS for educational purposes which they included in their (IACS) scabies photos gallery. We have chosen to remove the figure from this submission. 

Reviewer #1: Congratulations to the authors for coming up with this manuscript. As yet, there seems not to be any data on this topical issue in Ghanaian prisons. The Document is well written but will require some further corrections.

Abstract:

1. For the statement “and ascertain if the prevalence was high enough to warrant mass drug administration (MDA) with ivermectin…”, Consider deleting this as it is not an objective but a derived conclusion. Note that this study does not determine if MDA should be done.

Response: It is generally recognised that Mass Drug Administration (MDA) for scabies control is indicated if the community prevalence is more than 10% (Marks M, McVernon J, Engelman D, Kaldor J, Steer A. Insights from mathematical modelling on the proposed WHO 2030 goals for scabies. Gates Open Research. 2019;3:1542). Where such high prevalence rates (>10%) exist, treatment of individual cases and their contacts alone is considered not adequate for controlling scabies in such populations. Thus our aim included ascertaining if the prevalence was going to be high enough to warrant MDA with ivermectin. Our study revealed a prevalence of 65.8% which is greater than the 10% threshold recommended for initiating MDA. The section is therefore unchanged.

2. “79.2% of 96 individuals previously treated still demonstrated skin manifestations of scabies.”. When was this previous treatment? Was it before the study or a follow-up after the study? This must be indicated.

Response: We indicated in the methods section (line 171-173) that ‘’ Questions on treatment included whether participants had received any scabies treatment in the preceding months, what treatment was received if any and a description of how the treatment was used.’’ We also stated in the results section (line 223-224) that ‘’ At the time of assessment, 96 of the participants with scabies had recently received treatment in the past two months because of their skin problems.’’ We have revised the section in the abstract to now read: ‘79.2% of 96 individuals previously treated in the preceding two months still demonstrated skin manifestations of scabies.’ (line 58-60) 

3. “as it is likely that the criteria to start MDA for scabies are regularly met in prisons worldwide”: What is the basis for this statement?

Response: As indicated in the response to review question 1 above, the criteria for scabies MDA is when the prevalence exceeds 10% at which point treatment of affected individuals alone is insufficient for population control of the disease. Previous studies on MDA for scabies have been largely confined to island populations of the Pacific region (Romani L, et al. Mass Drug Administration for Scabies - 2 Years of Follow-up. The New England journal of medicine. 2019;381(2):186-7; Romani L, et al. Efficacy of mass drug administration with ivermectin for control of scabies and impetigo, with coadministration of azithromycin: a single-arm community intervention trial. The Lancet Infect Dis. 2019;19(5):510-8). We therefore consider it necessary to study the implementation of MDA in non-island settings including prisons. We have deleted the phrase ‘’ as it is likely that the criteria to start MDA for scabies are regularly met in prisons worldwide’’.

The section has been revised to now read: 

‘’The implementation of MDA and its effectiveness in the prison setting should be studied.’’

Background

1. “and ascertain if its prevalence was alarmingly high enough to warrant MDA with ivermectin in the prison.” : How is this to be determined?

Response: See response to question 1 above. 

Methods

1. “Every third prisoner within a cell who provided informed consent was enrolled”: As scabies is greatly determined by the space and environment one lives in, the sampling technique should be appropriately described to enable readers to determine if it is representative enough of the prison population.

Response: We have revised the section to now read: 

‘A convenience sampling technique was employed to identify and select participants for the study. We aimed to include one third of all inmates in the study. To achieve this, inmates from each cell or dormitory within the prison were made to stand in line and every third prisoner from who provided informed consent was enrolled.’ (line 146-149)

Results

1. Table 2 should rather come first. The contents of Table 1 are not mentioned in the first two paragraphs, after which Table 2 is mentioned.

Response: Table 1 describes characteristics of participants with and without scabies. Indeed, in the description of the results, we first describe the proportion of participants who had scabies (368/559) in Paragraph 2 of the results section and indicate that this information is shown in Table 1. Only after this do we describe the contents of Table 2. We have made no changes to the section.

2. The entirety of the contents of Table 3 is also in Table 1, and thus just a repetition. It should be deleted.

Response: Done as suggested.

3. The norm is that p-values should be exact if >=0.001 before cutting them short as < .001. I suggest you do just that to these p-values. A p-value of 0.04 is 10x that of 0.004!

Response: Done as suggested

4. In Table I for the rows: “Number of people per cell, median (IQR)” and “Time (median/IQR)”: There was no valid test to compare medians in the statistical analysis section, Is that what is being done here? If so, please put that in the statistical analysis section as well.

Response: We have added the sentence to the statistical analysis section:

‘Medians were compared using the Mann-Whitney test.’

5. Why is the p-value for the “Duration of stay (median/IQR) in the prison (months)” blank? Should it not be filled in?

Response: Done as suggested

6. Abbreviations such as “IACS” in the tables should be explained in the legend.

Response: Done as requested 

Reviewer #2: Major Comments:

1. Authors reported the scabies prevalence of 65.8% among inmates which is significantly higher. Authors are suggested not rely only on clinical diagnosis but also prefer confirmatory tests like dermoscopy, culturing, PCR etc. especially in such a high prevalence rate.

Response: We thank the reviewer for the comment. Dermoscopy allows for direct visualization of the scabies mites. A systematic review of diagnostic methods of scabies has shown low accuracy of dermoscopy in diagnosing scabies and clinical diagnosis remains an accepted practice in field studies [Leung V, Miller M. Detection of scabies: A systematic review of diagnostic methods. Can J Infect Dis Med Microbiol. 2011;22(4):143-6]. In addition, dermoscopy is not readily available in resource limited settings like ours where the study was conducted. Further, dermoscopy is more likely to be useful tool in settings with lower number of patients to screen and a lower pretest likelihood of scabies than in an outbreak setting. The criteria by the International Alliance for Control of Scabies (IACS) have been validated and shown to be useful for scabies diagnosis in varied field settings. Indeed, we are unaware of any recommendation regarding using PCR for scabies diagnosis in standard care; and culture may be performed for confirming superinfection when impetigo is present. In low resource settings where scabies prevalence is highest, tools like PCR and others mentioned by the reviewer are not readily available and clinical diagnosis using criteria like those proposed by IACS are deemed adequate.

2. Authors reported that a significant percentage (79.2%) of previously treated individuals still showed skin manifestations of scabies. This raises questions about treatment efficacy and possibility of reinfection and its potential factors. Authors are suggested to discuss the potential reasons behind this observation/treatment failure. It would be appropriate to include the exploring strategies for more effective treatment.

Response: We thank you for the comment. We previously included the requested information in the discussion section. We stated ‘In our study, 17% of participants had received treatment for their skin problems in the past 2 months. However, 79.2% of the individuals who were previously treated still exhibited skin manifestations of scabies; these may include individuals with post scabies itch, reinfections/ recurrences or non-response to treatment. There had been no treatment of the contacts of previously identified scabies cases. The lack of a systematic approach to the treatment of previously identified scabies cases and their contacts likely led to reinfection of individuals and ongoing spread within the prison.’ (lines 281-287).

Regarding strategies for effective scabies control, we included the sentences:

‘There are different strategies for scabies control. A control strategy of treating clinically affected persons and their contacts may provide temporary relief for the individuals but has limited success in reducing population prevalence in the longer term [26]. 

Mass drug administration (MDA) using topical permethrin or oral ivermectin offers an alternative approach for population control to substantially reduce the burden of scabies. Mass treatment of highly endemic communities with topical 5% permethrin has been reported to substantially reduce scabies prevalence [27-30]. In another study with permethrin, although the scabies prevalence remained unchanged, the prevalence of secondary infected scabies decreased from 3.7% to 1.5% representing a relative reduction of 59% [31]. 

Topical permethrin is expensive and requires application to the whole body. Other disadv

---

## [Decision Letter · Decision Letter 1]

29 Aug 2024

PONE-D-24-01158R1Burden of scabies in a Ghanaian penitentiaryPLOS ONE

Dear Dr. Amoako,

Thank you for submitting your manuscript to PLOS ONE. After careful consideration, we feel that it has merit but does not fully meet PLOS ONE’s publication criteria as it currently stands. Therefore, we invite you to submit a revised version of the manuscript that addresses the points raised during the review process.

We look forward to receiving your revised manuscript.

Kind regards,

Joanna Tindall, PhD

Staff Editor

PLOS ONE

Journal Requirements:

**Additional Editor Comments:**

The two reviewers have different views on the current status of your manuscript. Reviewer 2 holds concerns that their queries were not adequately addressed in the first round of revisions and that this prevents the work meeting PLOS ONE publication criteria. Please review the round 1 comments. along with any here, and clearly and fully address these points.

Reviewers' comments:

Reviewer's Responses to Questions

**Comments to the Author**

1. If the authors have adequately addressed your comments raised in a previous round of review and you feel that this manuscript is now acceptable for publication, you may indicate that here to bypass the “Comments to the Author” section, enter your conflict of interest statement in the “Confidential to Editor” section, and submit your "Accept" recommendation.

Reviewer #1: All comments have been addressed

Reviewer #2: (No Response)

2. Is the manuscript technically sound, and do the data support the conclusions?

Reviewer #1: Yes

Reviewer #2: Partly

3. Has the statistical analysis been performed appropriately and rigorously? 

Reviewer #1: Yes

Reviewer #2: No

4. Have the authors made all data underlying the findings in their manuscript fully available?

Reviewer #1: Yes

Reviewer #2: Yes

5. Is the manuscript presented in an intelligible fashion and written in standard English?

Reviewer #1: Yes

Reviewer #2: No

6. Review Comments to the Author

Reviewer #1: Congratulations to the authors for coming up with this revision. All the comments dealt with appropriately

Reviewer #2: Despite the revisions, significant flaws are still exist. Many of the previously raised concerns and comments have not been adequately addressed in the revised manuscript. As the persistence of these issues and the authors' inability to address them adequately, I regret to inform you that the manuscript cannot be accepted for publication in its current form. I encourage the authors to carefully consider and address the comments in any future resubmissions.

7. PLOS authors have the option to publish the peer review history of their article (what does this mean?). If published, this will include your full peer review and any attached files.

Reviewer #1: **Yes: **Samuel Blay Nguah

Reviewer #2: No

---

## [Author Response · Author response to Decision Letter 1]

9 Sep 2024

8th September 2024

Kumasi Centre for Collaborative Research

Kwame Nkrumah University of Science and Technology 

Kumasi, Ghana

The Editor

PLOS One Journal

Dear Sir,

Re: ‘Burden of scabies in a Ghanaian penitentiary’ [PONE-D-24-01158R1] - [EMID:dc590fc36ba29ccb]

My co-authors and I have taken note of the review comments and revised the manuscript as suggested. 

We wish to submit the revised manuscript for publication in your esteemed journal. In the attached reply, the reviewer questions are in blue font and our responses are in red font. 

We look forward to the next steps towards the publication of our manuscript.

Yours Sincerely,

Dr Yaw Ampem Amoako

Corresponding author

Point-by-point response to review comments

Journal Requirements:

Response: Done

Additional Editor Comments:

The two reviewers have different views on the current status of your manuscript. Reviewer 2 holds concerns that their queries were not adequately addressed in the first round of revisions and that this prevents the work meeting PLOS ONE publication criteria. Please review the round 1 comments. along with any here, and clearly and fully address these points.

Response: We thank the Editor and the reviewers for making time to review our manuscript. We have taken note of the comments and have worked to address the concerns. The manuscript has been revised according. 

Comments to the Author

1. If the authors have adequately addressed your comments raised in a previous round of review and you feel that this manuscript is now acceptable for publication, you may indicate that here to bypass the “Comments to the Author” section, enter your conflict of interest statement in the “Confidential to Editor” section, and submit your "Accept" recommendation.

Reviewer #1: All comments have been addressed

Reviewer #2: (No Response)

Response: Thank you.

2. Is the manuscript technically sound, and do the data support the conclusions?

Reviewer #1: Yes

Reviewer #2: Partly

Response: Thank you.

6. Review Comments to the Author

Reviewer #1: Congratulations to the authors for coming up with this revision. All the comments dealt with appropriately

Response: Thank you.

Reviewer #2: Despite the revisions, significant flaws are still exist. Many of the previously raised concerns and comments have not been adequately addressed in the revised manuscript. As the persistence of these issues and the authors' inability to address them adequately, I regret to inform you that the manuscript cannot be accepted for publication in its current form. I encourage the authors to carefully consider and address the comments in any future resubmissions.

Response: Thank you. We have taken note of the comments of the reviewer. We have revised the manuscript based on the comments from the first round of revisions as detailed below.

Reviewer #2: Major Comments:

1. Authors reported the scabies prevalence of 65.8% among inmates which is significantly higher. Authors are suggested not rely only on clinical diagnosis but also prefer confirmatory tests like dermoscopy, culturing, PCR etc. especially in such a high prevalence rate.

Response: We thank the reviewer for the comment. 

We used the consensus criteria for the clinical diagnosis of scabies established by the International Alliance for Control of Scabies (IACS) for scabies diagnosis in the current study. This criteria developed via a Delphi study, showed high agreement among experts (Engelman D, Fuller LC, Steer AC. Consensus criteria for the diagnosis of scabies: A Delphi study of international experts. PLoS Negl Trop Dis. 2018;12(5):e0006549; Engelman D, Yoshizumi J, Hay RJ, et al. The 2020 International Alliance for the Control of Scabies Consensus Criteria for the Diagnosis of Scabies. Br J Dermatol. 2020 Nov;183(5):808-820) and has been validated and shown subsequently to be useful for scabies diagnosis in varied field settings (Walker SL, Collinson S, Timothy J, et al. A community-based validation of the International Alliance for the Control of Scabies Consensus Criteria by expert and non-expert examiners in Liberia. PLoS Negl Trop Dis. 2020 Oct 5;14(10):e0008717; Matthews A, Le B, Amaral S, et al. Prevalence of scabies and impetigo in school-age children in Timor-Leste. Parasit Vectors. 2021 Mar 15;14(1):156; Amoako YA, Phillips RO, Arthur J, et al. A scabies outbreak in the North East Region of Ghana: The necessity for prompt intervention. PLoS Negl Trop Dis. 2020 Dec 22;14(12):e0008902). 

Furthermore, the IACS criteria is recommended for the diagnosis of scabies in implementation research and mapping projects, and for surveillance after control interventions (Engelman D, Fuller LC, Steer AC. Consensus criteria for the diagnosis of scabies: A Delphi study of international experts. PLoS Negl Trop Dis. 2018;12(5):e0006549).

While dermoscopy allows for direct visualization of the scabies mites, a systematic review of diagnostic methods of scabies reported low accuracy of dermoscopy in diagnosing scabies and clinical diagnosis remains an accepted practice in field studies (Leung V, Miller M. Detection of scabies: A systematic review of diagnostic methods. Can J Infect Dis Med Microbiol. 2011;22(4):143-6; Engelman D, Yoshizumi J, Hay RJ, et al. The 2020 International Alliance for the Control of Scabies Consensus Criteria for the Diagnosis of Scabies. Br J Dermatol. 2020 Nov;183(5):808-820). 

In addition, dermoscopy is not readily available in resource limited settings like ours where the study was conducted. Further, dermoscopy is more likely to be useful tool in settings with lower number of patients to screen and a lower pretest likelihood of scabies than in an outbreak setting. 

Indeed, we are unaware of any recommendation regarding using PCR for scabies diagnosis in standard care; and culture may be performed for confirming superinfection when impetigo is present. 

In low resource settings where scabies prevalence is highest, tools like PCR and others mentioned by the reviewer are not readily available and clinical diagnosis using criteria like those proposed by IACS have been shown in multiple studies to be adequate for implementation research and surveillance (Engelman D, Fuller LC, Steer AC. Consensus criteria for the diagnosis of scabies: A Delphi study of international experts. PLoS Negl Trop Dis. 2018;12(5):e0006549; Walker SL, Collinson S, Timothy J, et al. A community-based validation of the International Alliance for the Control of Scabies Consensus Criteria by expert and non-expert examiners in Liberia. PLoS Negl Trop Dis. 2020 Oct 5;14(10):e0008717; Matthews A, Le B, Amaral S, Arkell P, et al. Prevalence of scabies and impetigo in school-age children in Timor-Leste. Parasit Vectors. 2021 Mar 15;14(1):156).

We have also included in the limitations section (line 354-357) the following sentences:

‘We did not perform molecular based tests like PCR nor use tools like cultures for laboratory confirmation of fungal or bacterial infections. No skin biopsies were taken for histological examination. These tests are expensive and not routinely available to the population in which the current study was conducted.’ 

2. Authors reported that a significant percentage (79.2%) of previously treated individuals still showed skin manifestations of scabies. This raises questions about treatment efficacy and possibility of reinfection and its potential factors. Authors are suggested to discuss the potential reasons behind this observation/treatment failure. It would be appropriate to include the exploring strategies for more effective treatment.

Response: We thank you for the comment. We included the requested information in the discussion section. We state that: 

‘In our study, 17% of participants had received treatment for their skin problems in the 2 months preceding the assessment. However, 79.2% of the individuals who were previously treated still exhibited skin manifestations of scabies; these may include individuals with post scabies itch, reinfections/ recurrences or non-response to treatment/ treatment failure. There had been no treatment of the contacts of previously identified scabies cases. The lack of a systematic approach to the treatment of previously identified scabies cases and their contacts likely led to reinfection of individuals and ongoing spread within the prison.’ (lines 275-281). 

We have also added the following sentences to the discussion section (line 281-285).: ‘Additionally, the overcrowded living conditions, limited access to health care and inadequate Water, Sanitation and Hygiene (WASH) facilities may have facilitated scabies spread within the prison. There is the need to explore strategies for more effective control of scabies within this prison setting.’ 

Regarding strategies for effective scabies control, we included the sentences (lines 290-317):

‘There are different strategies for scabies control. A control strategy of treating clinically affected persons and their contacts may provide temporary relief for the individuals but has limited success in reducing population prevalence in the longer term [27]. 

Mass drug administration (MDA) using topical permethrin or oral ivermectin offers an alternative approach for population control to substantially reduce the burden of scabies. 

Mass treatment of highly endemic communities with topical 5% permethrin has been reported to substantially reduce scabies prevalence [28-31]. In another study with permethrin, although the scabies prevalence remained unchanged, the prevalence of secondary infected scabies decreased from 3.7% to 1.5% indicating a relative reduction of 59% [32]. 

Topical permethrin is expensive and requires application to the whole body. Other disadvantages of topical agents include inadequate application to local lesions, a lack of privacy to undertake whole-body application, insufficient facilities to rinse off the cream, poor compliance among contacts of scabies cases and discomfort due to topical treatment [33]. These factors constitute major barriers for effective use of topical therapy for population control of scabies, especially in the prison setting.

Ivermectin, as an oral therapy can offset some of the challenges associated with topical treatment and can potentially increase compliance resulting in better control of scabies in affected populations [8, 34]. Studies on MDA with ivermectin performed in island settings [15, 16, 35] reported a reduced prevalence of scabies. Compared to topical permethrin, oral ivermectin resulted in larger reductions in scabies prevalence at 12 and 24 months [15]. Recently, mass screening and community treatment (MSAT) of cases and their contacts with oral ivermectin delivered by community health workers has been shown to be superior to mass screening followed by usual care involving referral to clinic for topical treatment in controlling scabies in India [34]. While MSAT with ivermectin showed promising results and has relatively lower costs, the approach might miss persons with subclinical scabies making it less effective than MDA [34]. Furthermore, MSAT does not target people who work in the prison and may be contacts of scabies cases.’ 

3. Authors stated the (MDA with ivermectin) as a potential solution, authors are suggested to include the potential drawbacks of this approach. Authors may add alternative combination approaches for scabies control.

Response: Thank you for this comment. The section on approaches to population control of scabies reads (lines 290-325):

‘There are different strategies for scabies control. A control strategy of treating clinically affected persons and their contacts may provide temporary relief for the individuals but has limited success in reducing population prevalence in the longer term [27]. 

Mass drug administration (MDA) using topical permethrin or oral ivermectin offers an alternative approach for population control to substantially reduce the burden of scabies. Mass treatment of highly endemic communities with topical 5% permethrin has been reported to substantially reduce scabies prevalence [28-31]. In another study with permethrin, although the scabies prevalence remained unchanged, the prevalence of secondary infected scabies decreased from 3.7% to 1.5% indicating a relative reduction of 59% [32]. 

Topical permethrin is expensive and requires application to the whole body. Other disadvantages of topical agents include inadequate application to local lesions, a lack of privacy to undertake whole-body application, insufficient facilities to rinse off the cream, poor compliance among contacts of scabies cases and discomfort due to topical treatment [33]. These factors constitute major barriers for effective use of topical therapy for population control of scabies, especially in the prison setting.

Ivermectin, as an oral therapy can offset some of the challenges associated with topical treatment and can potentially increase compliance resulting in better control of scabies in affected populations [8, 34]. Studies on MDA with ivermectin performed in island settings [15, 16, 35] reported a reduced prevalence of scabies. Compared to topical permethrin, oral ivermectin resulted in larger reductions in scabies prevalence at 12 and 24 months [15]. Recently, mass screening and community treatment (MSAT) of cases and their contacts with oral ivermectin delivered by community health workers has been shown to be superior to mass screening followed by usual care involving referral to clinic for topical treatment in controlling scabies in India [34]. While MSAT with ivermectin showed promising results and has relatively lower costs, the approach might miss persons with subclinical scabies making it less effective than MDA [34]. Furthermore, MSAT does not target people who work in the prison and may be contacts of scabies cases.

We found a scabies prevalence in the prison which is much higher than the 10% threshold set for MDA [36]. In the current study, individuals with scabies were treated with topical benzyl benzoate as recommended in Ghana. However, the high prevalence and large number of contacts (entire prison population including staff and health workers) make MDA with ivermectin imperative. An urgent collaborative effort involving multiple stakeholders such as the Ghana Prisons Service, the Ghana Health Service, civil society and non-governmental organisations and donors is needed to mobilise adequate doses of ivermectin to undertake an MDA in this institutionalized setting.’

We have revised the section on the drawbacks of ivermectin MDA in the discussion and it now reads as follows (line 326-340): 

‘Ivermectin is contraindicated in children < 5 years old, persons with severe concurrent illness, individuals with hypersensitivity to components of the drug, and pregnancy. Inadequate community engagement, lack of political support and high costs are potential barriers that may pose challenges for MDA implementation [37, 38]. Furthermore, due to its lack of ovicidal action, a second dose of ivermectin is usually recommended after 7-14 days for individual treatment, to kill newly hatched mites. Administrati

---

## [Decision Letter · Decision Letter 2]

1 Oct 2024

Burden of scabies in a Ghanaian penitentiary

PONE-D-24-01158R2

Dear Dr. Amoako,

We’re pleased to inform you that your manuscript has been judged scientifically suitable for publication and will be formally accepted for publication once it meets all outstanding technical requirements.

Kind regards,

Jorge Cervantes

Academic Editor

PLOS ONE

Additional Editor Comments (optional):

After the series of revisions I am recommending accepting this manuscript.

Although one of the Reviewers still selected Reject (Reviewer 2), I consider that the authors responses were satisfactory. 

Reviewers' comments:

Reviewer's Responses to Questions

**Comments to the Author**

1. If the authors have adequately addressed your comments raised in a previous round of review and you feel that this manuscript is now acceptable for publication, you may indicate that here to bypass the “Comments to the Author” section, enter your conflict of interest statement in the “Confidential to Editor” section, and submit your "Accept" recommendation.

Reviewer #1: All comments have been addressed

Reviewer #2: (No Response)

2. Is the manuscript technically sound, and do the data support the conclusions?

Reviewer #1: Yes

Reviewer #2: Partly

3. Has the statistical analysis been performed appropriately and rigorously? 

Reviewer #1: Yes

Reviewer #2: Yes

4. Have the authors made all data underlying the findings in their manuscript fully available?

Reviewer #1: Yes

Reviewer #2: Yes

5. Is the manuscript presented in an intelligible fashion and written in standard English?

Reviewer #1: Yes

Reviewer #2: No

6. Review Comments to the Author

Reviewer #1: The authors have addressed the comments made in the previous review. I congratulate them on completing this.

Reviewer #2: Authors stated in manuscript that "We did not perform molecular based tests like PCR nor

use tools like cultures for laboratory confirmation of fungal or bacterial infections. No skin

biopsies were taken for histological examination. These tests are expensive and not routinely

available to the population in which the current study was conducted". These tests are really very important in term of the justification of the findings and accuracy. The article has many technical issue which is not fit for the publication in PLOS ONE. Authors may reconsider some other journal.

7. PLOS authors have the option to publish the peer review history of their article (what does this mean?). If published, this will include your full peer review and any attached files.

Reviewer #1: No

Reviewer #2: No

---

## [Editor Report · Acceptance letter]

3 Oct 2024

PONE-D-24-01158R2 

PLOS ONE

Dear Dr. Amoako, 

I'm pleased to inform you that your manuscript has been deemed suitable for publication in PLOS ONE. Congratulations! Your manuscript is now being handed over to our production team.

Kind regards, 

on behalf of

Dr. Jorge Cervantes 

Academic Editor

PLOS ONE